# Fibroblast Subsets in Intestinal Homeostasis, Carcinogenesis, Tumor Progression, and Metastasis

**DOI:** 10.3390/cancers13020183

**Published:** 2021-01-07

**Authors:** Hao Dang, Tom J. Harryvan, Lukas J. A. C. Hawinkels

**Affiliations:** Department of Gastroenterology and Hepatology, Leiden University Medical Center, Albinusdreef 2, 2333 ZA Leiden, The Netherlands; h.dang@lumc.nl (H.D.); t.j.harrijvan@lumc.nl (T.J.H.)

**Keywords:** colorectal cancer, tumor stage, adenoma–carcinoma sequence, cancer-associated fibroblast

## Abstract

**Simple Summary:**

Colorectal cancer often develops via the adenoma–carcinoma sequence, a process which is accompanied by (epi) genetic alterations in epithelial cells and gradual phenotypic changes in fibroblast populations. Recent studies have made it clear that these fibroblast populations which, in the context of invasive cancers are termed cancer-associated fibroblasts (CAFs), play an important role in intestinal tumor progression. This review provides an overview on the emerging role of fibroblasts in various stages of colorectal cancer development, ranging from adenoma initiation to metastatic spread of tumor cells. As fibroblasts show considerable heterogeneity in subsets and phenotypes during cancer development, a better functional understanding of stage-specific (alterations in) fibroblast/CAF populations is key to increase the effectiveness of fibroblast-based prognosticators and therapies.

**Abstract:**

In intestinal homeostasis, continuous renewal of the epithelium is crucial to withstand the plethora of stimuli which can damage the structural integrity of the intestines. Fibroblasts contribute to this renewal by facilitating epithelial cell differentiation as well as providing the structural framework in which epithelial cells can regenerate. Upon dysregulation of intestinal homeostasis, (pre-) malignant neoplasms develop, a process which is accompanied by (epi) genetic alterations in epithelial cells as well as phenotypic changes in fibroblast populations. In the context of invasive carcinomas, these fibroblast populations are termed cancer-associated fibroblasts (CAFs). CAFs are the most abundant cell type in the tumor microenvironment of colorectal cancer (CRC) and consist of various functionally heterogeneous subsets which can promote or restrain cancer progression. Although most previous research has focused on the biology of epithelial cells, accumulating evidence shows that certain fibroblast subsets can also importantly contribute to tumor initiation and progression, thereby possibly providing avenues for improvement of clinical care for CRC patients. In this review, we summarized the current literature on the emerging role of fibroblasts in various stages of CRC development, ranging from adenoma initiation to the metastatic spread of cancer cells. In addition, we highlighted translational and therapeutic perspectives of fibroblasts in the different stages of intestinal tumor progression.

## 1. Introduction and Definitions

Colorectal cancer (CRC) is the most commonly diagnosed malignancy in the gastrointestinal tract, constituting almost two million new cases and one million deaths per year worldwide [1,2]. The majority of CRC cases develop via the adenoma–carcinoma sequence [3]. This process is accompanied by sequential accumulation of (epi) genetic alterations in epithelial cells that lead to the formation of benign precursor lesions called adenomas [4]. Some of these adenomas eventually progress into invasive carcinomas. Although most previous research has focused on the biology of epithelial cells, it has become clear that the tumor microenvironment (TME), also known as the tumor stroma, plays an important role in CRC initiation and progression as well [5,6]. For instance, a high stromal content at the invasive front of CRCs is strongly correlated to an increased risk of CRC-related death [7,8]. Moreover, several studies have demonstrated that the prognostic value of the recently developed Consensus Molecular Subtypes (CMS) classification system for CRC [9] can be mainly attributed to genes expressed by stromal cells, rather than tumor cells [10,11,12]. Of all the CMS categories, the mesenchymal or stromal CRC subtype (CMS4) is associated with the worst survival outcomes [9], thereby underlining the significant involvement of the TME in tumor progression.

The tumor stroma of CRC consists of several cell types (e.g., immune cells and endothelial cells), of which the cancer-associated fibroblasts (CAFs) are most abundant. CAFs are defined as fibroblasts surrounding malignant tumor cells [13], and consist of various heterogeneous subsets which can exert both tumor-promoting and -suppressing functions [5,6,13,14]. Many studies have shown that CAFs can make an important contribution to CRC progression, making them a promising target for improving therapeutic strategies for CRC [15]. However, a major challenge hampering progress in the field of CAF research is the lack of precision of fibroblast-specific markers (i.e., markers which can identify all the different fibroblast subsets and which are also not expressed in any other cell type) [13]. This has led to inconsistent definitions and nomenclature across studies on (cancer-associated) fibroblasts [16]. For our review, we have adopted the recommendations of a 2020 consensus statement on fibroblast research [13]. Fibroblasts are defined as spindle-shaped, elongated cells with slender cytoplasmic processes, which do not express lineage markers for endothelial (e.g., cluster of differentiation 31 (CD31)), epithelial (e.g., CD326), or immune cells (e.g., CD45) [17]. Fibroblasts with a highly contractile phenotype which are positive for alpha smooth muscle actin (α-SMA) will be referred to as myofibroblasts [18]. All fibroblasts (regardless of cellular origin) found within and surrounding an invasive or metastatic carcinoma will be referred to as CAFs [13]. Expression of other commonly used markers for (cancer-associated) fibroblasts such as fibroblast activation protein (FAP), platelet-derived growth factor receptors alpha (PDGFR-α) and beta (PDGFR-β), CD90, and alpha-1 type I collagen (COL1A1), is not included in our definitions but will be used to characterize and describe fibroblast subsets whenever reported. This is because not all fibroblasts express these markers, and not all studies use the same markers to define (cancer-associated) fibroblasts [5,13].

CAFs can have several different cellular origins, including non-fibroblastic cell populations (e.g., epithelial and endothelial cells) and remote circulating cells (bone marrow derived mesenchymal stem cells) [5,6,13]. However, the many studies which have described gradual changes occurring in the fibroblast compartment during cancer development [19,20,21] suggest that the majority of CAFs initially originate from tissue-resident fibroblasts [13]. In intestinal homeostasis, resident fibroblasts are usually in a quiescent state and can be activated as part of a wound healing response [22], which is initiated upon damage to the intestinal epithelium caused by mechanical (peristalsis and fecal stream) or erosive (invasive bacteria and chemical agents) stimuli. Activated fibroblasts can acquire a highly contractile phenotype, produce increased amounts of extracellular matrix (ECM), and secrete factors which stimulate tissue repair and regeneration [5]. The activation of fibroblasts during the wound healing response shows high resemblance to the formation of tumor stroma [23]. Under physiological conditions, activated fibroblasts are reverted to their original state or undergo apoptosis after the structural integrity of the tissue has been restored. However, in cancers, which have also been described as “wounds that do not heal” [23], it is believed that this resolution phase is disrupted—as the tumor progresses into more advanced stages, local fibroblast populations are undergoing gradual changes and eventually acquire a CAF-phenotype [19,24,25]. Currently, the exact sequential changes are not well understood, because longitudinal sampling of the same tumor throughout disease progression is often impossible.

In this review, we summarized the current literature on the emerging role of fibroblasts in various stages of CRC development, ranging from adenoma initiation to the metastatic spread of tumor cells. As considerable heterogeneity in fibroblast phenotypes can exist along the continuum of cancer development [6], special attention was given to currently known fibroblast subsets and their (possible) roles in tumor progression. We also highlighted translational and therapeutic perspectives of fibroblasts in the different stages of intestinal tumorigenesis.

## 2. Fibroblasts in Intestinal Homeostasis

First, to provide some context on (cancer-associated) fibroblasts in neoplastic disease, we will briefly discuss the characteristics and functions of intestinal (myo) fibroblasts in physiology. As summarized in several excellent reviews [16,17,18,26], fibroblasts are phenotypically heterogeneous and contribute to intestinal homeostasis via multiple mechanisms (Figure 1A). They are mainly known for their role in ECM remodeling, which is essential for maintaining the structural integrity of the intestinal mucosa [17,22,27]. However, intestinal fibroblasts can also regulate proliferation and differentiation of epithelial (stem) cells via secretion of wingless-related integration site (Wnt) ligands and bone morphogenetic proteins (BMP) antagonists [26,28,29,30]. In addition, fibroblasts play important roles in intestinal immune cell homeostasis by directly interacting with immune cells or secreting various inflammatory cytokines, such as interleukin-6 (IL-6) or C-C motif chemokine ligand 2 (CCL2) [17,31].

### 2.1. Fibroblast Subsets in Normal Intestinal Mucosa

For decades, it has been notoriously difficult to define phenotypically distinct fibroblast subsets and attribute specific functions to them, because they often express overlapping marker genes and proteins. However, the recent introduction of single-cell transcriptomics [32] has provided a powerful analysis tool for unbiased delineation and comprehensive characterization of fibroblast subsets. In 2018, the colonic mesenchyme of healthy humans was analyzed using single-cell RNA sequencing (scRNAseq) for the first time [33]. Unbiased clustering analysis revealed five fibroblast subsets with distinct expression profiles. One of these subsets was identified as myofibroblasts, while the other four expressed much lower levels of myofibroblast genes (*α-SMA*, *myosin heavy chain 11 (MYH11)*, and *gamma-enteric smooth muscle actin (ACTG2)*) and were termed S1–S4. The S1 subset was characterized by enrichment for ECM-related genes, and in particular for non-fibrillar collagens (e.g., *COL14A1* and *COL15A*) and elastic fibers (*fibronectin 1 (FN1)* and *fibulin 2 (FBLN2)*). The S2 subset was enriched for key constituents of the epithelial basement membrane (*COL4A5* and *COL4A6*), and also showed high expression of several ligands of the transforming growth factor β (TGF-β) superfamily (*BMP2* and *BMP5*) and Wnt pathway (*Wnt5a* and *frizzled related protein (FRZB)*). The S3 subset (marker genes: e.g., *osteoglycin (OGN)* and *gelsolin (GSN)*) was enriched for genes involved in organization of supramolecular fibers and extracellular clusters, and the S4 subset mainly showed enrichment for immune-related processes (e.g., T-cell activation, antigen processing and presentation). Examples of S4 marker genes were *IL32*, *CD74*, and *interferon regulatory factor 8 (IRF8)*. Although these analyses have shed some light on the heterogeneous intestinal fibroblast compartment, functional attributes of the identified subsets largely remain to be elucidated. 

### 2.2. Fibroblasts and Intestinal Adenoma Formation

Enhanced adenoma initiation has been linked to alterations in several intestinal fibroblast-related factors (Figure 1B). For example, Nik et al. reported that *adenomatous polyposis coli (APC)* mutant mice developed significantly more intestinal adenomas upon loss of one allele for *forkhead box F2 (FOXF2)*, a transcription factor which is mainly expressed in intestinal fibroblasts [34]. Complete knockout of *nod-like receptor pyrin domain-containing protein 6 (NLRP6)*, which is primarily expressed by colonic myofibroblasts, resulted in significantly more colorectal tumors than control mice in a mouse model of colitis-associated tumorigenesis (azoxymethane-dextran sulfate sodium (AOM/DSS) model) [35]. Similar results were observed in the AOM/DSS model upon fibroblast-specific (*COL1A2*-driven Cre expression) loss of *inhibitor of nuclear factor kappa-B kinase subunit beta (IKKβ)* [36] and loss of *tumor progression locus 2 (TPL2)* in myofibroblasts (*COL6*-Cre) [37]. In contrast, complete knockout of *periostin*, an ECM protein, which in the intestinal mucosa appears to be mainly derived from stromal fibroblasts, considerably reduced the number of colorectal tumors in both AOM/DSS-treated and *APC*-mutant mice [38]. Collectively, these findings suggest that alterations in the normal intestinal fibroblast compartment could importantly contribute to the development of neoplastic disease.

Several mechanisms have been proposed via which fibroblast alterations may lead to enhanced intestinal adenoma formation. For instance, both *IKKβ*-deficient fibroblasts and *TPL2*-deficient myofibroblasts were found to secrete elevated levels of hepatocyte growth factor (HGF) [36,37], an important cytokine involved in tumor development and progression [39,40,41], and pharmacological inhibition of the HGF receptor (MET) could revert the phenotype caused by (myo)fibroblast-specific deletion of these targets [36,37]. These results indicate that intestinal fibroblast alterations may promote adenoma formation via aberrant HGF/MET signaling. Another signaling pathway which may be affected by alterations in fibroblasts includes Wnt signaling. *FOXF2*^+/−^ fibroblasts expressed lower levels of the Wnt antagonist *secreted frizzled-related protein 1 (SFRP1)* [34], thereby possibly contributing to overactive Wnt signaling and subsequent hyperproliferation of intestinal stem cells [42,43,44]. Besides, there are also some signs that alterations in normal fibroblast populations may lead to the formation of a microenvironment which facilitates adenoma formation. For example, myofibroblast-specific inactivation (*FOXL1*-Cre) of the *BMP receptor 1a (BMPR1A/ALK3)* in mice led to expansion of the fibroblast compartment and remodeling of the intestinal microenvironment, which resulted in a higher number of polyps compared to control mice [45]. Further support for the hypothesis of microenvironmental remodeling prior to adenoma formation was reported by Guo et al., who showed that non-neoplastic normal intestinal mucosa from patients with advanced adenomas contained a significantly higher proportion of senescent fibroblasts than normal mucosa from individuals without any neoplastic intestinal disease [46]. These senescent fibroblasts could promote proliferation of epithelial cells in vitro via secretion of Growth Differentiation Factor 15 (GDF15) [46], thereby possibly contributing to adenoma formation. Altogether, these studies indicate that alterations in normal intestinal fibroblasts may indirectly contribute to adenoma formation via altered paracrine signaling or remodeling of the intestinal microenvironment. 

In vivo evidence of direct fibroblast-mediated adenoma initiation is quite scarce. Only recently, it was demonstrated that a pericryptal fibroblast subset expressing *prostaglandin-endoperoxide synthase 2* (*PTGS2*, also known as *cyclooxygenase-2 (COX-2)*) could directly initiate the formation of adenomas in *APC*-mutant mice (fibroblast-specific targeting using *COL6*-driven Cre expression) [47]. Mechanistically, this was found to occur via the paracrine prostaglandin E_2_ (PGE_2_)—PGE_2_ receptor 4 (PTGER4)—Yes-associated protein (YAP) signaling axis [47]. The COX-2 fibroblast-driven adenoma-initiating pathway could be an important underlying mechanism behind the CRC-preventative effect of COX-2 inhibitors in humans [48], and might also provide a promising alternative (i.e., PTGER4 inhibition [49]) for chemoprevention of CRC given the substantial side effects of COX-inhibiting drugs [48]. In the healthy human mucosa, *COX-2* expression was mainly observed in S1 fibroblasts, and in particular in the S1 subset with high expression levels of *fibroblast-growth factor 2 (FGF2)* and *vascular cell adhesion molecule 1 (VCAM1)*, and low levels of *PDGFR-α* expression [33]. However, COX-2 expressing fibroblasts are not unique to one of the major fibroblast subtypes in the normal mucosa, as *COX-2* is to a lesser extent also expressed in S2–S4 fibroblasts. Besides, the amount of this adenoma-initiating fibroblast subset appears to be regulated via several mechanisms. For instance, stromal Indian hedgehog (IHH) signaling may be important for maintaining the population of COX-2 expressing fibroblasts, as loss of IHH signaling in APC-mutant mice considerably reduced both the number of intestinal adenomas and COX-2 expression levels [50]. A similar decrease in adenoma count and COX-2 expression was reported when trametinib, a small-molecule mitogen-activated protein kinase kinase (MEK) inhibitor, was administered to *APC* mutant mice [51], suggesting that the COX-2 expressing subset could also be maintained by MEK signaling. In another study, it was shown that COX-2 expression in fibroblasts may be induced by pericellular hypoxia caused by densely populated epithelial cells [52]. These findings may indicate the existence of a reciprocal loop in which (fibroblast-mediated) epithelial hyperproliferation may escalate towards adenoma formation by increasing the number of adenoma-promoting COX-2 expressing fibroblasts. It is currently unclear whether or not such an expansion also occurs before adenoma formation in humans. 

In summary, fibroblasts in normal intestinal mucosa display considerable phenotypic heterogeneity and actively contribute to intestinal homeostasis via several mechanisms. These include remodeling of the ECM as well as interaction with epithelial and immune cells. Alterations in normal intestinal fibroblasts (e.g., loss of *IKKβ* or *BMPR1A*) may indirectly lead to enhanced adenoma formation via altered paracrine signaling or remodeling of the intestinal microenvironment. Evidence of direct fibroblast-mediated adenoma formation in vivo has only been provided for COX-2 expressing fibroblasts, an adenoma-promoting subset which can already be found in healthy human intestine. The abundance of this fibroblast subset appears to be regulated via various pathways, such as IHH or MEK signaling. Although these findings emphasize the involvement of intestinal fibroblasts in the initiation of neoplastic disease, it largely remains to be elucidated how the composition of the heterogeneous fibroblast compartment in the human intestine changes prior to adenoma formation, and to what extent these changes in (relative abundance of) fibroblast subsets contribute to adenoma initiation.

## 3. Fibroblasts in Intestinal Carcinogenesis

### 3.1. Adenomas

There are multiple reports of tissue fibrosis (i.e., a chronic wound healing response as a result of unabated tissue injury) and expansion of certain fibroblast subsets occurring in precursor lesions of solid tumors [53,54,55,56,57]. Currently, it remains a point of discussion whether these stromal changes accelerate or provide protection from progression to invasive carcinomas. It is generally assumed that the pre-malignant stroma to a certain degree contributes to malignant progression [5,23], since multiple studies have linked tissue fibrosis (which is characterized by activation of fibroblasts) to an increased risk of developing malignancies in various organs [56,57,58,59]. However, for pancreatic precursor lesions it has been shown that depletion of all α-SMA-expressing myofibroblasts increases the number of invasive carcinomas [60], suggesting that the pre-malignant stroma may also play an important role in preventing malignant transformation [61]. For intestinal adenomas, neither of these theories has been supported by functional studies yet. It has been shown that serrated adenomas can progress towards the mesenchymal CMS4 subtype via aberrant TGF-β signaling [62], and that TGF-β-activated CAFs can promote CRC formation and progression [10,63]. However, direct evidence for the TGF-β-mediated tumor-promoting function of fibroblasts in serrated adenomas has not been provided yet. Likewise, Glentis et al. demonstrated that colorectal CAFs can stimulate the invasion of cancer cells through the basement membrane [64], but it is questionable whether these fibroblasts, which were mostly isolated from advanced CRCs (≥stage III), can already be found in premalignant lesions. 

To date, the fibroblast compartment of intestinal adenomas has not been comprehensively characterized, and current knowledge exclusively comes from microscopic and immunohistochemical analyses. The earliest observations of changes occurring in the fibroblast compartment of premalignant lesions date back to 1971, when it was reported that the pericryptal fibroblast sheath in adenomas showed continued fibroblast division and failure of morphological maturation compared to the physiological situation [65]. Years later, several studies showed that this fibroblast sheath underwent stage-specific changes during intestinal carcinogenesis (Figure 2). For example, the number of pericryptal myofibroblasts appeared to decrease in the adenoma, intramucosal and submucosally invasive carcinoma sequence [66,67]. A similar decrease was observed in the sequence of polypoid, flat and centrally depressed adenomas (Figure 3) [67,68], possibly explaining the observation that submucosal invasion occurs more often in non-polypoid tumors (and in particular in the depressed ones) compared to polypoid tumors [69]. Depressed adenomas also displayed lower levels of COX-2 expression in pericryptal myofibroblasts than flat or polypoid lesions [70], and expression levels decreased even further when submucosal invasion was present [71]. This corroborates with the finding that COX-2 expressing fibroblasts are only required for adenoma formation and not for adenoma progression [47]. A comparable decrease was also observed for periostin expression in pericryptal fibroblasts [72], possibly serving as another example of the dispensability of adenoma formation-related factors in the process of malignant transformation. However, it remains to be elucidated whether the decrease in (subsets of) pericryptal myofibroblasts actually initiates tumor cell invasion or only occurs as a secondary event caused by other invasion-promoting processes. 

There are also several markers which are increasingly being expressed by (myo) fibroblasts during the adenoma–carcinoma sequence and, thus, might indicate subsets involved in malignant progression. These markers include heparanase [73], proline-glutamic acid-leukine rich protein 1 (PELP1) [74], estrogen receptor beta (ERβ) [75], urokinase-type plasminogen activator [76], CD10 [77], and podoplanin [78]. Interestingly, high expression levels of podoplanin in ≥stage II CRCs were reported to be associated with a favorable prognosis [79,80], and co-culture experiments also demonstrated that *podoplanin* knockdown in CAFs could enhance CRC cell invasion [79]. These findings suggest that the increase in podoplanin expression might reflect a compensatory expansion of an invasion-suppressing fibroblast subset during CRC development. On the contrary, CD10-expressing fibroblasts seem to be important drivers of malignant transformation. For example, functional studies showed that the invasive potential of CRC stem cells could be significantly enhanced by CD10-positive CAFs (derived from ≥stage II CRCs) [81], a subset which has also been shown to exert tumor-promoting functions in breast cancer [82]. Other signs of an invasion-promoting role of CD10-expressing fibroblasts include the observation that the highest levels of CD10 expression in CRCs were found at the invasive front [77,83]. Lastly, fibroblasts expressing certain matrix metalloproteinases (MMPs) might also promote tumor cell invasion. For instance, stromal fibroblasts appear to express increasing levels of MMP-1 and MMP-9 during CRC development [84]. Remarkably, such an increase was not observed in tumors of patients with Lynch syndrome [84], which might explain why these tumors have a less invasive potential than sporadic tumors. There are also some other MMPs which are increasingly expressed by stromal fibroblasts in the adenoma–carcinoma sequence, such as MMP-15 and MMP-19 [85]. Again, it remains unclear whether the aforementioned associations represent a causal relationship with tumor invasion or not. 

In short, immunohistochemical studies suggest that the fibroblast compartment in intestinal adenomas changes dynamically during progression towards invasive cancer. These changes include expansion of certain fibroblasts (e.g., CD10-, podoplanin- or specific MMP-expressing fibroblasts), and a decrease in the number of pericryptal myofibroblasts and expression levels of associated proteins (COX-2, periostin). Functional studies are required to evaluate whether or not these changes in fibroblast composition mechanistically contribute to malignant progression of intestinal adenomas.

### 3.2. Early-Invasive Colorectal Cancers

As soon as tumor cells have invaded through the basement membrane and muscularis mucosae into the submucosa, they are able to further invade the intestinal wall and metastasize to other organs [86]. Although submucosally invasive cancers (T1 CRCs) are by definition in the earliest stage of wall invasiveness, around 6–12% of tumors have already metastasized to lymph nodes or other organs [87,88,89] and require surgical treatment. However, current risk stratification models are far from accurate, resulting in >80–90% unnecessary surgical resections for T1 CRCs [90,91,92]. To optimize risk stratification, a better understanding of the biology of these tumors is needed. Recently, it was reported that stromal/CAF expression patterns may be useful for predicting patient prognosis in T1 CRCs [93,94,95], suggesting that T1 CRC CAFs might also be importantly involved in early cancer progression. So far however, evidence for this hypothesis is lacking. Unpublished data from our research group showed that compared to matched normal fibroblasts, T1 CRC-derived CAFs had distinct gene expression profiles with around 400 differentially expressed genes (mainly related to ECM remodeling). It would be interesting to see whether these differentially expressed genes mark out CAF subsets in T1 CRC which play important roles in tumor progression and/or could serve as biomarkers of aggressive disease in T1 CRC patients. 

## 4. Fibroblasts in Advanced CRC Progression and Metastasis

Compared to the role of fibroblasts in early CRC development, much more is known about CAFs in advanced stage CRCs and the wide range of tumor progression-related functions that they can exert (already summarized in several excellent reviews [6,17,96,97]). However, most of these findings have not yet been translated into applications with clinically relevant efficacy, mainly due to the considerable heterogeneity in CAF subsets and phenotypes in advanced CRCs [98,99,100,101,102]. To tackle this issue, Li et al. performed scRNAseq on human primary CRCs and found that CAFs seem to cluster into two major CAF types termed CAF-A (marker genes: e.g., *MMP-2*, *FAP*, and *decorin*) and CAF-B (marker genes: e.g., *α-SMA*, *transgelin* and *platelet-derived growth factor subunit A (PDGFA)*) [103]. Although it remains unclear how these CAF clusters are functionally involved in CRC progression, findings from clinical trials suggest that they are not exclusively composed of tumor-promoting or tumor-suppressing CAFs. For instance, therapeutic targeting of FAP-expressing CAFs (i.e., the CAF-A subset) does not appear to significantly affect clinical outcomes of patients with advanced CRC [97,104,105]. These results suggest that a more detailed, function-based subclassification of CAFs is required. 

In contrast to the scarce literature on tumor-suppressing functions of CAFs, there are quite some studies which have related various tumor-promoting functions to specific markers expressed by certain CAFs found in human CRCs (Table 1). However, as most of these CAFs have not been extensively characterized, it often remains unclear how they relate to the two major CAF clusters and whether they represent phenotypically distinct subsets or overlap with other tumor-promoting or -suppressing CAFs. In this section, we provided an overview on (the markers of) these possible CAF subsets, stratified per tumor-promoting function which they have been linked to.

### 4.1. Tumor Growth, Invasion, and TME Remodeling

Numerous studies have shown that CAFs can importantly contribute to cancer cell proliferation and invasion [5,6,13,14]. In addition, they are also able to affect to tumor progression via remodeling of the ECM or regulation of tumor angiogenesis [106,107,108]. In advanced CRCs, tumor cell proliferation can be promoted by CAFs which express hydrogen peroxide-inducible clone-5 (HIC-5) [109] or Snail-1 [110], or CAFs which secrete FGF-1, FGF-3 [111], or exosomal circular RNA SLC7A6 [112]. Interestingly, some of these CAFs were also found to promote tumor progression via other mechanisms such as enhancing angiogenesis [111] or CRC cell invasion [110,112,113]. Another example of a “multifunctional subset” includes Wnt2-expressing CAFs, which can promote CRC cell proliferation and migration [114,115], facilitate CRC invasion via ECM remodeling [115], and increase tumor angiogenesis by secreting several pro-angiogenic factors (e.g., angiopoietin-2 (ANG2), placental growth factor (PGF)) [116]. Besides, high levels of Wnt2-expressing CAFs were also associated with an increased risk of cancer metastasis and recurrence in advanced CRCs [114,115]. A comparable association with worse patient prognosis was found for high expression levels of microRNA-21 [117], a factor which is predominantly expressed in CAFs and marks a CAF population which could support CRC cell proliferation and invasion [118,119].

### 4.2. Therapeutic Resistance and Immune Regulation

It is well known that CAFs can also facilitate cancer progression by promoting tumor cell resistance to cytotoxic therapies [120,121]. Several studies have demonstrated that certain CAFs in human CRCs are also able to do so. For example, radiation-induced apoptosis in CRC cells can be reduced by CAFs which express microRNA-31 [122] or microRNA-93–5p [123]. Moreover, CAFs which express long non-coding RNA (lncRNA) H19 [124] and colorectal cancer-associated lncRNA (CCAL) [125], can promote CRC cell stemness and chemoresistance via exosomal transfer of these lncRNAs to tumor cells. TGF-β2 secreting CAFs have also been shown to enhance chemoresistance in a paracrine manner by upregulating glioma-associated oncogene family zinc finger 2 (GLI2) expression in CRC stem cells [126]. These CAFs may serve as an important target for prognostic applications, since high expression levels of TGF-β2 were strongly associated with an increased risk of relapse in chemotherapy-treated CRC patients [126]. In addition, therapeutic targeting of these CAFs may also become feasible in the (near) future, with several TGF-β inhibitors already being tested in clinical trials [127,128,129,130]. Next to supporting therapy resistance, certain CAFs in advanced CRCs can actively contribute to immune evasion of tumor cells (e.g., via aberrant TGF-β signaling [131]) and inhibition of anti-tumor immune responses [132,133,134,135]. Li et al. reported that CAFs expressing C-X-C motif chemokine 5 (CXCL5) can promote expression of programmed death-ligand 1 (PD-L1) [136], an important suppressor of T-cell activity [137]. The fact that CAFs are able to produce a defense against T-cells could have important implications for immunotherapies against CRC [132,133,134,135]. Moreover, CD70-expressing CAFs were shown to increase the survival of naturally occurring regulatory T-cells [138], which are key mediators of immunosuppression [139]. CD73-expressing CAFs were also able to enhance immune suppression via adenosine receptor 2A signaling [140]. Notably, the abundance of CD70- and CD73-expressing CAFs was significantly associated with a worse prognosis in CRC patients [138,140,141], thereby emphasizing the involvement of immunomodulatory CAFs in CRC progression.

### 4.3. Metastasis

Currently, metastatic disease remains the major cause of death in CRC patients [153]. CAFs can play an important role in mediating CRC metastasis, as summarized by Tommelein et al. [154]. Recent work from our group identified several metastasis-promoting CAFs present at the invasive front of primary CRCs, which were also significantly correlated with poor metastasis-free patient survival. These include CAFs which express endoglin [150] or BMP2 [151]. Interestingly, BMP2 expression in CAFs seemed to be regulated by tumor necrosis factor-related apoptosis-inducing ligand (TRAIL), a cytokine which is overexpressed in CRC cells with a deficiency for mothers against decapentaplegic homolog 4 (SMAD4) [151]. These findings provide further support for the idea that the interaction between CRC cells (with a certain mutational status) and CAFs could eventually progress to a tumor-promoting reciprocal loop [155,156,157,158]. Other examples of metastasis-promoting CAF populations include IL-11 secreting CAFs, which could activate apoptosis-suppressing programs in metastatic tumor cells [63], and stanniocalcin-1 (STC1) expressing CAFs [152]. Lastly, next to CAFs in the primary tumor, several studies have suggested that CAFs at the metastatic site, which could originate from resident fibroblasts in remote organs [159,160,161] or CAFs co-travelling with tumor cells [162,163], can also contribute to CRC metastasis. For instance, resident liver and lung fibroblasts can induce formation of the pre-metastatic niche via (primary) tumor cell-driven upregulation of pro-inflammatory cytokines such as IL-6 and IL-8 [159,160,161]. Studies on circulating CAFs originating from primary CRCs have not been conducted yet, but evidence from other cancer types suggests that circulating CAFs may be importantly involved in tumor metastasis and, thus, may serve as useful prognosticators [164]. For breast and prostate cancer, it has been shown that circulating CAFs are predominantly found in patients with metastatic disease, and rarely in patients with localized cancer [165,166]. 

In brief, various tumor-promoting functions (e.g., enhancing tumor invasion, therapy resistance, or metastasis) have been linked to markers expressed by certain CAF populations in advanced CRC. Some of them also seem to serve as promising targets for prognostic or therapeutic applications. However, clinical translation of these findings should be cautiously considered, since it is yet unknown whether these CAF markers demarcate functionally distinct subsets or show overlap with other tumor-promoting or -suppressing CAFs. To enable more precise identification and targeting of “unfavorable” CAFs in advanced CRC, attention needs to be paid to comprehensive characterization and reporting of the subsets being investigated in CAF studies.

## 5. Conclusions

Over the past few decades, it has become clear that (cancer-associated) fibroblasts are importantly involved in intestinal tumor initiation, progression, and metastasis. Yet, we are still in the infancy of translating these findings into useful clinical applications for CRC patients. This is mainly due to the considerable phenotypic heterogeneity that the intestinal fibroblast compartment exhibits throughout cancer development. As a result, strategies which involve targeting or normalization of “unfavorable” fibroblast/CAF subsets are often not specific enough, resulting in a suboptimal clinical efficacy or on-target side effects. Recent advances in single-cell analysis techniques (e.g., scRNAseq) have provided powerful tools for tackling this heterogeneity by allowing precise identification of distinct fibroblast subsets, but most of the identified subsets have not yet been functionally investigated. Contrarily, multiple tumor-related functions have been attributed to specific markers expressed by certain fibroblasts (and in particular CAFs in advanced stage CRCs), but extensive characterization of these populations is often lacking or inadequately reported. To resolve these issues, we think that future research should focus on connecting findings from characterization and functional studies with each other. In our opinion, the recently published study on COX-2 fibroblast-driven adenoma formation [47] could serve as an example how this can be achieved, with extensive in vitro and in vivo data on a specific fibroblast population being linked to subsets identified in scRNAseq analyses. Another point which deserves special attention is the context which the studied fibroblast subsets originate from, given the stage-dependent variability of fibroblasts during CRC development. This variability is for example reflected in the aforementioned immunohistochemical studies on fibroblast alterations during malignant transformation of adenomas. To allow proper interpretation and translation of study findings, extensive documentation on fibroblast metadata (e.g., histology and stage of the originating tumor and spatial location of investigated fibroblasts in the tissue) is therefore of the uttermost importance. With all of the aforementioned things in place, we believe that research into (cancer-associated) fibroblasts may ultimately provide valuable tools which contribute to optimizing clinical care for patients with intestinal tumors.

## Figures and Tables

**Figure 1 cancers-13-00183-f001:**
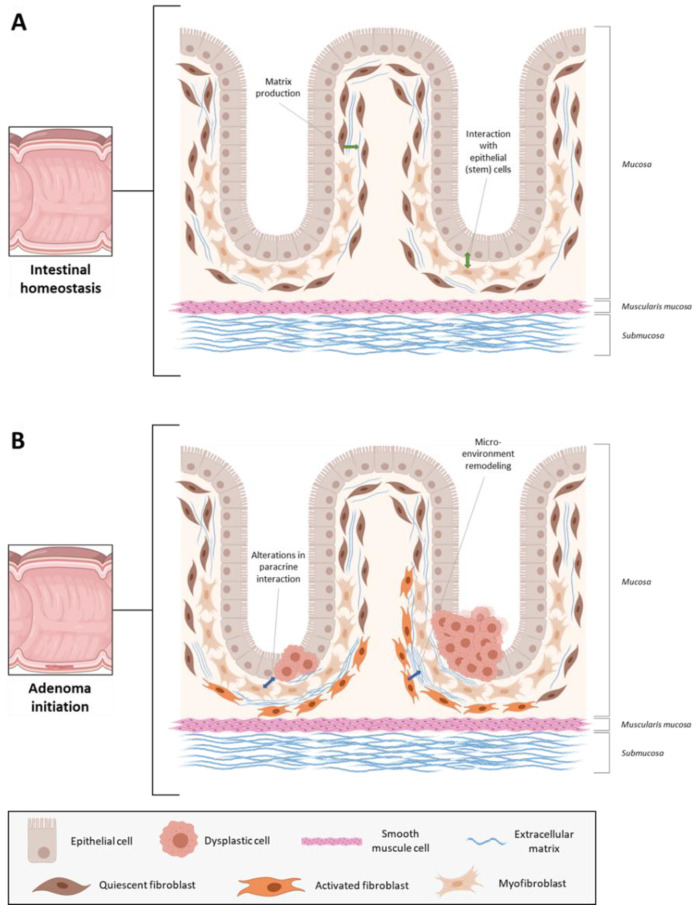
Fibroblasts in (**A**) intestinal homeostasis and (**B**) adenoma initiation. (Created with BioRender.com).

**Figure 2 cancers-13-00183-f002:**
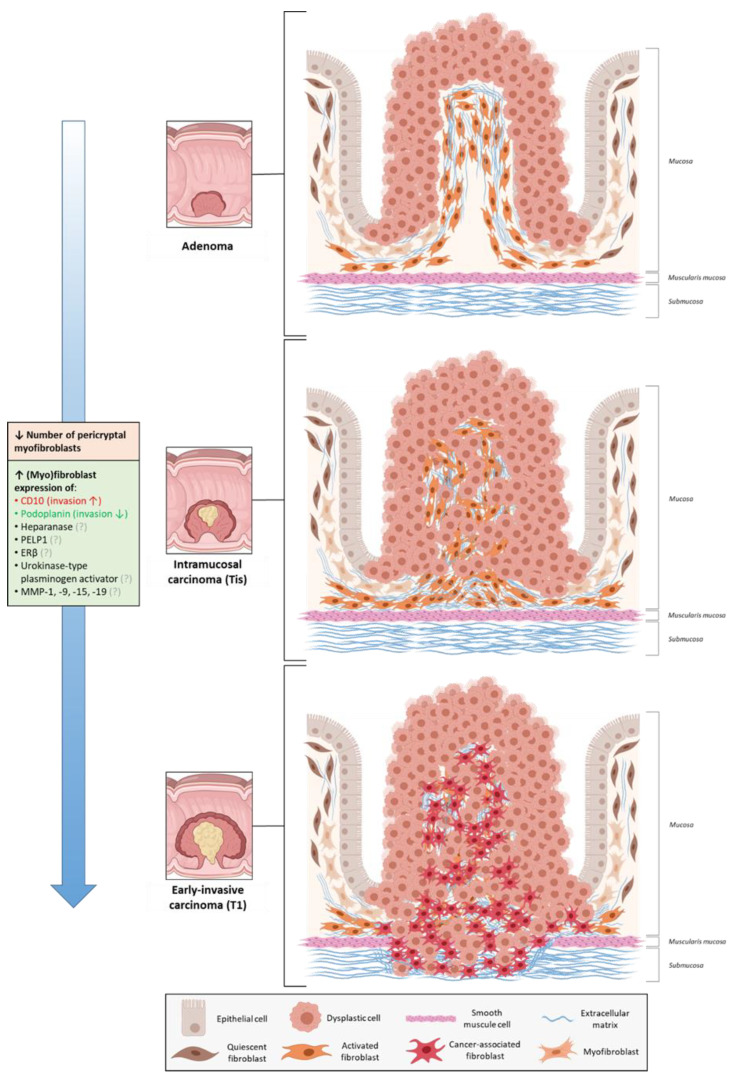
Fibroblasts during malignant progression of intestinal adenomas. CD, cluster of differentiation; PELP1, proline-glutamic acid-leukine rich protein 1 (PELP1); ERβ, estrogen receptor beta; MMP, matrix metalloproteinase. (Created with BioRender.com).

**Figure 3 cancers-13-00183-f003:**
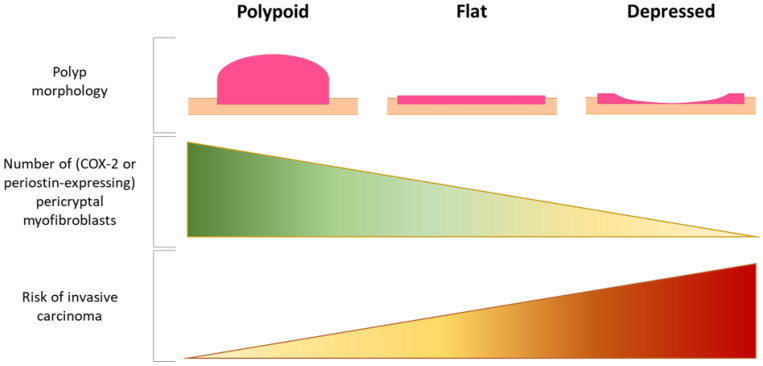
Relation between polyp morphology, pericryptal myofibroblasts, and risk of invasive cancer. (Created with BioRender.com).

**Table 1 cancers-13-00183-t001:** Examples of tumor-related functions which have been linked to factors expressed or secreted by certain cancer-associated fibroblast (CAF) populations in human advanced colorectal cancers (CRCs).

CAF Metadata			Role in CRC Progression						Clinical Relevance
Name of Factor	Characteristics of CAFs Expressing or Secreting This Factor	Characteristics of Tumors Which the CAFs Originated From	CRC Proliferation	CRC Invasion/Migration	Angiogenesis	Therapy Resistance	Immune Evasion	Organ Metastasis	Association High Expression Levels in CAFs & Poor Clinical Outcomes
Exosomal circular RNA SLC7A6 [112]	-	Primary tumor; no neo-adjuvant treatment	↑	↑	-	-	-	-	Yes [112]
HIC-5 [109]	Co-localization with α-SMA	Primary tumor	↑	-	-	-	-	-	-
COX-2 [142,143]	-	Primary tumor	↑	-	-	-	-	-	-
Snail-1 [110]	Co-localization with α-SMA and FAP	Primary tumor	↑	↑	-	-	-	-	Yes [144]
FGF-1 [111]	Vimentin-positive	Primary tumor	↑	↑	↑	-	-	-	-
FGF-3 [111]	Vimentin-positive	Primary tumor	↑	↑	↑	-	-	-	-
Wnt2 [114,115,116]	-	Primary tumor; rectum; stage II-IV; well/moderate differentiation [114]. Not described in [115,116]	↑	↑	↑	-	-	-	Yes [114,115]
miRNA-21 [118,119]	-	Primary tumor; rectum & sigmoid; stage II-III; well/moderate differentiation; microsatellite stable; no neo-adjuvant treatment	↑	↑	-	-	-	↑	Yes [117]
CD10 [81]	-	Primary tumor; rectum and colon; stage II-III; well/moderate differentiation	↑	↑	-	-	-	-	-
CLEC3B [145]	-	Primary tumor	-	↑	-	-	-	-	Yes [145] ^1^
Podoplanin [79]	-	-	-	↓	-	-	-	-	No [79,80] ^2^
SPARC [146]	-	Primary tumor	-	↑	-	-	-	-	Yes [146]
IGF-2 [147]	-	Primary tumor	=	↑	-	-	-	↑	Yes [147]
RAB31 [148]	Co-localization with α-SMA and vimentin	Primary tumor	=	↑	-	-	-	-	Yes [148]
CCBE1 [149]	Co-localization with α-SMA	Primary tumor	-	-	↑	-	-	-	Yes [149]
miRNA-31 [122]	-	Primary tumor; no neo-adjuvant treatment	-	-	-	↑	-	-	-
miRNA-93–5p [123]	-	Primary tumor	-	-	-	↑	-	-	-
CRC-associated lncRNA [125]	FAP-positive, co-localization with α-SMA	Primary tumor	-	-	-	↑	-	-	-
lncRNA H19 [124]	-	Primary tumor	-	-	-	↑	-	-	-
TGF-β2 [126]	-	Primary tumor	-	-	-	↑	-	-	Yes [126]
CXCL5 [136]	-	Primary tumor	-	-	-	-	↑	-	-
CD70 [138]	Co-localization with α-SMA and FAP	Primary tumor	-	↑	-	-	↑	-	Yes [138,141]
CD73 [140]	Co-localization with α-SMA	Primary tumor	-	-	-	-	↑	-	Yes [140]
Endoglin [150]	Co-localization with α-SMA	Primary tumor	-	-	-	-	-	↑	Yes [150]
BMP2 [151]	-	Primary tumor	-	↑	-	-	-	↑	Yes [151] ^3^
IL-11 [63]	FAP-positive	Primary tumor	-	-	-	-	-	↑	-
STC1 [152]	-	Primary tumor	-	↑	-	-	-	↑	-

^1^ Combined with high expression of α-SMA. ^2^ High expression levels of podoplanin in CAFs were associated with favorable clinical outcomes in CRC patients. ^3^ Only in patients with SMAD4-deficient tumors. CAF: cancer-associated fibroblast, CRC: colorectal cancer, ECM: extracellular matrix, α-SMA: alpha smooth muscle actin, FAP: fibroblast activation protein, COX-2: cyclooxigenase-2, HIC-5: hydrogen peroxide-inducible clone-5, FGF: fibroblast growth factor, miRNA: microRNA, CD: cluster of differentiation, RAB31: Ras-related protein RAB-31, CLEC3B: C-type lectin domain family 3 member B, SPARC: secreted protein acidic and rich in cysteine, IGF-2: insulin-like growth factor 2, CCBE1: collagen and calcium-binding epidermal growth factor domain 1, lncRNA: long non-coding RNA, TGF- β2: transforming growth factor-β2, CXCL5: C-X-C motif chemokine 5, BMP2: bone morphogenetic protein 2, IL-11: interleukin 11, STC1: stanniocalcin-1. ↑ indicates a tumor-promoting role, ↓ indicates a tumor-suppressing role, = indicates a neutral role (i.e., neither tumor-promoting nor tumor-suppressing).

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
