# Peer review of "Fibroblast Subsets in Intestinal Homeostasis, Carcinogenesis, Tumor Progression, and Metastasis"

_cancers, 2021, doi:10.3390/cancers13020183_

Round 1

Reviewer 1 Report

Line 7

Detection of soluble clinical  biomarkers is an important objective in order to obtain cheaper and incruent diagnostic methods that could contribute to humanize colorectal cancer patient management. This could be stated in the “Simple Summary” section. It could be something like this: “Colorectal cancer progression shows gradual changes in fibroblasts,  leading  to possible histochemical and soluble biomarkers that could be easily detected and could facilitate prognosis and treatment in patients.”

Line 73

Although fibroblast activation protein (FAP) is not expressed by all fibroblasts in colorectal cancer microenvironment, it is a useful biomarker in order to predict survival and metastasis. Consequently, the sentence “This is because not all fibroblasts express these markers, and the use or reporting of these markers is often inconsistent across studies “ is not adequate, at least in reference to FAP.

Line 90

The intestinal “wound that do not heal” is accompanied by physiological changes in fibroblast function that probably lead to soluble biomarkers. It can not be discarded that detection of theese markers in body fluids could alert on the possibility of future colorectal cancer. This could be of special interest in order to test the effects of diets or chemical/pharmacological agents on intestinal stroma, as well as to protect patients from cancer risk. This could be included in text.

Line 322

The paragraph “Unpublished data from our research group showed that compared to matched normal fibroblasts, T1 CRC-derived CAFs had distinct gene expression profiles with around 400 differentially expressed genes. Overrepresented annotations among these genes were mainly related to ECM remodeling. Preliminary results suggest that when compared to CAFs derived from more advanced stage tumors, some of these ECM-genes also seem to display opposing expression levels in T1 CRC-derived CAFs” is out of place in a review paper. Unpublished data can not be contrasted. Preliminary results should not be presented in this paper.

Line 341

The sentence “For instance, simply targeting FAP-expressing CAFs (i.e. the CAF-A subset) does not appear to significantly affect clinical outcomes of patients with advanced CRC [95, 101, 102].” contradicts other papers that could be cited and discussed, such as the following:

"Altered expression of fibroblast activation protein-α (FAP) in coloectal adenoma-carcinoma sequence and in lymph node and liver metastases". Jon Danel Solano-Iturri, Maider Beitia, Peio Errarte, Julio Calvete-Candenas, María C. Etxezarraga, Alberto Loizate, Enrique Echevarria, Iker Badiola, Gorka Larrinaga. Aging (Albany NY) 2020 Jun 15; 12(11): 10337–10358.

"High Expression of FAP in Colorectal Cancer Is Associated With Angiogenesis and Immunoregulation Processes". Mairene Coto-Llerena, Caner Ercan, Venkatesh Kancherla, Stephanie Taha-Mehlitz, Serenella Eppenberger-Castori, Savas D. Soysal, Charlotte K. Y. Ng, Martin Bolli, Markus von Flüe, Guillaume P. Nicolas, Luigi M. Terracciano, Melpomeni Fani, Salvatore Piscuoglio. Front Oncol. 2020; 10: 979.

Line 387  

With regard to the paragraph “Next to supporting therapy resistance, certain CAFs in advanced CRCs can actively contribute to immune evasion of tumor cells and inhibition of anti-tumor immune responses [127, 128].CAFs expressing C-X-C motif chemokine 5 (CXCL5) can promote expression of programmed death-ligand 1 (PD-L1)[129], an important suppressor of T-cell activity [130]”,  the  role of  PD-L1 requires more extension and some references in this paper. The fact that CAFs can produce a defense against T killers was an important finding, and its implications in metastasis and therapy are very important, thus deserving a place in this review.

Line 396

Table 1 “Examples of tumor-related functions which have been linked to factors expressed or secreted by certain CAF populations in human advanced CRCs” is difficult to understand and should be improved in its presentation.

Line 425

With respect to the paragraph “…several studies have suggested that CAFs at the metastatic site, which could originate from resident fibroblasts inremote organs [145-147] or CAFs co-travelling with tumor cells [148, 149], can also contribute to CRC metastasis”, this is an important point that could be developed in the paper. The fascinating possible role of fibroblasts arriving to intact distant organs and contributing to metastasis open the possibility of sampling other organs in colorectal cancer patients, searching for signals of defense, even before the reception of cancer cells. Moreover, new biomarkers for metastasis could be discovered, both soluble or in the surface of fibroblasts present in plasma, with the aid of flow cytometry. Some opinions about these points, even speculative, could be discussed in this paper.

Reviewer 2 Report

This is an interesting and comprehensive review of the role of CAFs in colorectal cancer (CRC). The authors have extensively reviewed the literature and discuss several aspects about CAFs, including the need to better characterize CAF populations in all stages of CRC, the possibility that they participate in the initiation of CRC and their changes during cancer progression. I would like to add just a few suggestions listed below.

1. It would be really nice to have a graphical abstract or a final figure capturing the main points of the review.

2. I think some interesting studies could complete this review. For instance, the heterogeneity of CAFs has been suggested to have prognostic value and non-tumor fibroblast also present important changes (doi 10.1016/j.molonc.2014.04.006), the role of TGFb could be further discussed as it has been suggested that it is involved in immune evasion (doi 10.1038/nature25492), and chemoresistance to 5FU by CAFs has also been reported (10.18632/oncotarget.11121).

3. I do not see any studies regarding serrated CRC. Is there any literature available discussing the role of CAFs in this type of cancer? Evidence suggests that TGFb is involved in its development (doi 10.15252/emmm.201606184), and CAFs are the main contributors to TGFb production.
